# Effectiveness of Lifestyle Modification vs. Therapeutic, Preventative Strategies for Reducing Cardiovascular Risk in Primary Prevention—A Cohort Study

**DOI:** 10.3390/jcm11030688

**Published:** 2022-01-28

**Authors:** Małgorzata Chlabicz, Jacek Jamiołkowski, Wojciech Łaguna, Marlena Dubatówka, Paweł Sowa, Magda Łapińska, Anna Szpakowicz, Natalia Zieleniewska, Magdalena Zalewska, Andrzej Raczkowski, Karol A. Kamiński

**Affiliations:** 1Department of Population Medicine and Lifestyle Diseases Prevention, Medical University of Białystok, 15-259 Białystok, Poland; mchlabicz@op.pl (M.C.); jacek909@wp.pl (J.J.); marlena.dubatowka@umb.edu.pl (M.D.); mailtosowa@gmail.com (P.S.); magda.lapinska@umb.edu.pl (M.Ł.); drobeknatalia@gmail.com (N.Z.); mzalewska@o2.pl (M.Z.); andrzej.raczkowski@umb.edu.pl (A.R.); 2Department of Invasive Cardiology, Medical University of Białystok, 15-259 Białystok, Poland; 3Faculty of Computer Science, Bialystok University of Technology, 15-259 Białystok, Poland; wojciech.laguna@gmai.com; 4Department of Cardiology, Medical University of Białystok, 15-259 Białystok, Poland; akodi@poczta.onet.pl

**Keywords:** cardiovascular risk, lifestyle modification, population studies

## Abstract

Background: Cardiovascular diseases (CVD) are still the leading cause of death in developed countries. The aim of this study was to calculate the potential for CV risk reduction when using three different prevention strategies to evaluate the effect of primary prevention. Methods: A total of 931 individuals aged 20–79 years old from the Bialystok PLUS Study were analyzed. The study population was divided into CV risk classes. The Systematic Coronary Risk Estimation (SCORE), Framingham Risk Score (FRS), and LIFE-CVD were used to assess CV risk. The optimal prevention strategy assumed the attainment of therapeutic goals according to the European guidelines. The moderate strategy assumed therapeutic goals in participants with increased risk factors: a reduction in systolic blood pressure by 10 mmHg when it was above 140 mmHg, a reduction in total cholesterol by 25% when it was above 190 mg/dL, and a reduction in body mass index below 30. The minimal prevention strategy assumed that CV risk would be lowered by lifestyle modifications. The greatest CV risk reduction was achieved in the optimal model and then in the minimal model, and the lowest risk reduction was achieved in the moderate model, e.g., using the optimal model of prevention (Model 1). In the total population, we achieved a reduction of −1.74% in the 10-year risk of CVD death (SCORE) in relation to the baseline model, a −0.85% reduction when using the moderate prevention model (Model 2), and a −1.11% reduction when using the minimal prevention model (Model 3). However, in the low CV risk class, the best model was the minimal one (risk reduction of −0.72%), which showed even better results than the optimal one (reduction of −0.69%) using the FRS. Conclusion: A strategy based on lifestyle modifications in a population without established CVD could be more effective than the moderate strategy used in the present study. Moreover, applying a minimal strategy to the low CV risk class population may even be beneficial for an optimal model.

## 1. Introduction

Cardiovascular diseases (CVD) are still the leading diseases in the European population [1] despite the fact that we have guidelines for CVD prevention, lifestyle, and the management of risk factors [2,3]. The latest European primary prevention survey showed that a large number of individuals with high CVD risk have inadequate control of lipids, blood pressure (BP), and diabetes and maintain unhealthy lifestyles [4]. Furthermore, multiple European reports of secondary prevention have revealed that a large number of patients with CVD maintain unhealthy lifestyles in terms of their diet, smoking, and sedentary behavior and that they did not achieve their low-density lipoprotein cholesterol (LDL-C), BP, and glucose targets [5,6]. Thus, it is imperative to reduce CVD risk by improving preventive programs.

Many studies show that a healthy lifestyle and the control of disorders such as hypertension (AH), hypercholesterolemia, and excess body weight prevent CVD events. Nevertheless, many of these disorders may go undiagnosed and therefore untreated. Accordingly, special attention should be given to those without a CVD diagnosis and who are at increased risk of CVD who consider themselves healthy and who may be reluctant to undergo diagnostic and therapeutic procedures. Therefore, in this study, we focused on assessing CV risk in people without a known CVD, most of whom were unaware of their increased CV risk. For this purpose, we used various validated calculators to assess CV risk in primary prevention [7,8]. The Systematic Coronary Risk Estimation—Polish version (Pol-SCORE) was used to assess the 10-year risk of fatal CV based on the following risk factors: age, gender, smoking, systolic blood pressure (BPs), and total cholesterol (TC) for individuals aged 40–70 [9,10]. The Framingham Risk Score (FRS) was used to predict the 10-year risk of developing the first CVD event (coronary death, myocardial infarction (MI), coronary insufficiency, angina, ischemic stroke, hemorrhagic stroke, transient ischemic attack, peripheral artery disease, or heart failure) using scores for lipids or body mass index (BMI) based on the following factors: age, diabetes, smoking, treated and untreated BPs, TC, high-density lipoprotein cholesterol (HDL-c), or BMI-replacing lipids [11]. The lifetime-perspective model for individualizing cardiovascular disease prevention strategies in apparently healthy people (LIFE-CVD) estimates the probability of survival free of heart attack or stroke; a 10-year risk of MI, stroke, or CV death; and a lifetime risk of MI, stroke, or CV death using the following factors: age, gender, smoking, geographic region, diabetes, parental history of MI prior to age 60, BPs, BMI, TC, HDL-c, and low-density lipoprotein cholesterol (LDL-c) [12].

We designed three different prevention models. In the optimal model (therapeutic prevention strategy (Model 1)), we ideally assumed the attainment of therapeutic goals in patients with present CV risk factors: successful BPs reduction below 130 mmHg; LDL-C reduction below 116 mg/dL (3.0 mmol/L) in the low CV risk class, below 100 mg/dL (2.6 mmol/L) in the moderate CV risk class, below 70 mg/dL (1.8 mmol/L) in the high CV risk class, and below 55 mg/dL (1.4 mmol/L) in the very high CV risk class; TC level reduction below 156 mg/dL (4.0 mmol/L); BMI reduction below 25; and smoking cessation. In the moderate therapeutic prevention strategy (Model 2), we assumed more realistic therapeutic goals: successful BPs lowering by 10 mmHg when over 140 mmHg; LDL-C reduction by 25% when over 116 mg/dL (3.0 mmol/L) in the low CV risk class, over 100 mg/dL (2.6 mmol/L) in the moderate CV risk class, over 70 mg/dL (1.8 mmol/L) in the high CV risk class, and over 55 mg/dL (1.4 mmol/L) in the very high CV risk class; TC level reduction by 25% when over 190 mg/dL (4.9 mmol/L); and BMI reduction to below 30. In the minimal prevention strategy (a population-based approach (Model 3)), we assumed risk factors modifications that are attainable via lifestyle modifications, namely the lowering the following values in the analyzed population: TC and LDL-C levels by 10%, BPs by 5 mmHg, and BMI by 5%, accompanied by smoking cessation.

The aim of this study was to calculate the potential for CV risk reduction using the three different prevention strategies outlined above to evaluate the effect of primary prevention using validated and well-known scales across a total population as well as by CV risk categories.

Calculating risk reduction by using different degrees of prevention intensity can help local authorities to improve their prevention programs. The use of real-time CV risk data is a powerful factor for the optimization of evidence-based policy. This data-driven approach can enable responsible people to choose the right health strategies that can lead to better health for all.

## 2. Materials and Methods

The Bialystok PLUS Study was conducted in 2017–2020 on a sample of Bialystok residents aged 20–79 years old. Bialystok is a city in the eastern part of Poland and has a population of 297,500 inhabitants.

### 2.1. Recruitment of Bialystok PLUS Study Participants

Each year (in the middle of the year, after June 30th, when demography statistics by the Main Statistical Office in Poland are calculated), we received a pseudonymized list of Bialystok citizens from the Municipal Office in Bialystok. The dataset was limited to people aged 20–79 years old, and categories based on gender and 5-year intervals (20–24, 25–29, etc.) were assigned, providing a total of 24 subcategories. We randomly sampled citizens from each subcategory separately, in such a number that allowed us to obtain a similar proportion distribution similar to that in the city’s population. After sampling, the identifiers of the selected citizens were sent back to the Municipal Office, and we received their names and addresses in order to contact them. The sampled citizens were invited to participate in the study through a letter and were encouraged to contact us by phone or email to schedule a visit. We sent second and even third invitation letters after a period of time for those who did not respond. We randomly sampled a number of citizens to be examined for the next year. There were no exclusion criteria; however, for pregnant women, some examinations were not carried out in certain subpopulations (e.g., OGTT in diabetics).

### 2.2. Data Collection

Data collection was conducted by trained research staff. At the time of study entry, a detailed medical history was collected from each patient using questionnaires. The comprehensive assessment was performed as described previously [7].

### 2.3. Ethical Issues

Ethical approval for this study was provided by the Ethics Committee of the Medical University of Bialystok (Poland) on 31 March 2016 (approval number: R-I-002/108/2016). The study was conducted in accordance with the Declaration of Helsinki, and all participants gave written informed consent.

### 2.4. Division into CV Risk Classes

According to the “2019 ESC/EAS guidelines for the management of dyslipidemias: lipid modification to reduce cardiovascular risk” [2] recommendations, the study population was divided into CV risk classes. A detailed description of the subdivision is presented elsewhere [8].

### 2.5. Designed Prevention Models

We designed three different prevention models, all of which are described above and are shown in Figure 1.

### 2.6. Calculators for the Assessment of CV Risk in Primary Prevention

The Systematic Coronary Risk Estimation (SCORE) was calculated, participants who were pre-qualified in the high and very high CV risk classes according to the aforementioned recommendations were excluded [2]. We used the Pol-SCORE system because it was recalibrated in Poland [9,10]. According to the original risk stratification in the ESC guidelines, participants with previously diagnosed CVD (myocardial infarction—MI, ischemic heart disease—IHD, stroke, transient ischemic attack—TIA, peripheral arterial disease—PAD, significant plaque on carotid ultrasound >50%), previously diagnosed DM or DM diagnosed at the time of study entry, moderate or severe chronic kidney diseases (CKD) at the time of study entry, markedly elevated single risk factor (TC > 310 mg/dL, LDL-c > 190 mg/dL, BP ≥ 180/110 mmHg) at the time of study entry, and individuals aged younger than 40 years old or older than 70 years old were excluded from the Pol-SCORE calculation, as they present a higher CV risk than that calculated according to the algorithm.

The Cardiovascular Disease Framingham Heart Study (FRS) predicted a 10-year risk of developing the first CVD event (coronary heart disease, stroke, peripheral artery disease, or heart failure) using scores for BMI or lipids based on the following factors: age, smoking, diabetes, treated and untreated BPs, TC, high-density lipoprotein cholesterol (HDL-C), or lipids replacing BMI [11]. Participants with previously diagnosed CVD (myocardial infarction—MI, ischemic heart disease—IHD, stroke, transient ischemic attack—TIA, peripheral arterial disease—PAD, significant plaque on carotid ultrasound >50%) and those who were younger than 30 years old or older than 74 years old were excluded from further analysis.

To assess lifetime risk, the LIFE-CVD was used. This model calculates a 10-year risk of myocardial infarction (MI), stroke, or CV death; lifetime risk of MI, stroke, or CV death; or the probability of survival-free heart attack or stroke using the following factors: age, gender, smoking, geographic region, diabetes, parental history of MI prior to age 60, BPs, BMI, TC, HDL-C, and LDL-C [12]. Participants with previously diagnosed CVD (MI, IHD, stroke, TIA, peripheral arterial disease, significant plaque on carotid ultrasound >50%), and who were younger than 45 years old or older than 79 years old were excluded from this analysis.

Patients in whom a certain risk assessment method could not be applied were excluded only from the analysis based on that particular method. Basic CV risk and the CV risk in the assumed models were estimated according to the Pol-SCORE [9], FRS [11], and LIFE-CVD [12,13] calculators. All of the analyses were also performed in the investigated population, which was divided according to CV risk classes using the latest 2019 ESC/EAS guidelines [2], with 931 people being included in the analyzed sample. A total of 560 met the age requirement, and 465 satisfied of the Pol-SCORE calculation conditions [9]. Predicting the 10-year risk for the development of the first CVD event using FRS scores, we analyzed 830 people from the sample who met the age requirements [11]. Overall, 721 people from the sample had sufficient data and satisfied all of the conditions [11] for Framingham lipid and BMI-based model calculation. To estimate the risk of the first CVD event within 10 years and over one’s lifetime (LIFE-CVD), we analyzed 531 people from the sample who met the age requirements, and 469 satisfied all of the conditions for LIFE-CVD calculation [12,13].

### 2.7. The Estimation of the Number of the Local Inhabitants

Estimations of the number of the local inhabitants who may die (Pol-SCORE) due to CVD within 10 years; who may develop their first CVD event within 10 years (FRS); or who may develop their first MI, stroke, or CV death within 10 years and within their lifetime (LIFE-CVD) were also performed. The scores that were used to estimate the risk are only applicable to primary prevention, that is, in people without previously diagnosed CVD. We obtained the frequency of CVD in our study population (931 participants), and based on this, we calculated the prevalence of CVD in the total population of the city using 5-year strata separately in the male and female participants. In 2018, the local population aged 20–79 years was equal to 208,303: 112,545 females and 95,758 males. After applying the aforementioned conversion method, using the 5-year strata, it was estimated that there were 196,184 inhabitants without CVD in the age group reflected in our study.

Applying the age criterion on the population (40 ≤ age ≤ 70) resulted in 104,998 people being able to participate. Calculating the Pol-SCORE risk for each person in the sample allowed us to estimate the number of people from the total population who would survive within 10 years after applying a particular prevention strategy. The expected number of deceased individuals was estimated based on the binomial distribution, where each person from the sample represented people from the population in the same age–gender category. We grouped the sample and population into age–gender categories, where each category contained people of the same gender arranged in five-year intervals.

The estimation of the number of people who would not develop their first CVD event (coronary heart disease, stroke, peripheral artery disease, or heart failure) within 10 years (FRS) was determined after applying a particular prevention strategy and applying the Framingham age criterion on the local population (30 ≤ age ≤ 74) and resulted in 158,183 people. Calculating the Framingham risk for each group allowed us to calculate the number of people who not develop their first CVD event within 10 years after applying prevention strategies. We grouped the sample and population into age–gender categories, with each category containing people of the same gender arranged in five-year intervals.

Applying the age criterion for LIFE-CVD on the local population (45 ≤ age ≤79) resulted in 98,086 people. Calculating the LIVE-CVD 10-year and lifetime risk for each person in the sample allowed us to estimate the number of people from the population who would not develop their first CVD event within 10 years or in their lifetime after the application of a particular prevention strategy. The expected number of deceased was estimated based on the binomial distribution, where each person from the sample represents people from the population in the same age–gender category. We grouped the sample and population into age–gender categories, with each category containing people of the same gender arranged in five-year intervals.

### 2.8. Statistical Analysis

Descriptive statistics for the quantitative variables were presented as means and 95% confidence intervals (95% CI) and as the counts and frequencies for the qualitative variables. The IBM SPSS Statistics 27.0 statistical software (Armonk, NY, USA) was used for all of the calculations.

## 3. Results

A total of 2449 residents were randomly selected from the mayor’s office database and were invited to participate in the study. A total of966 residents responded and were examined (Figure 2). Due to incomplete data, 35 people were excluded from further analysis. Overall, 931 individuals were included in the research group, out of which 63 (6.8%) participants had established CVD, 275 (29.6%) had a history of AH, 71 (7.6%) had a history of diabetes mellitus (DM), and 186 (20.1%) smoked cigarettes.

The mean age of the study population was 49.1 ± 15.5 years old, and 43.2% of the participants were male. The characteristics of the study population are presented in Table 1, and detailed data are elsewhere [8]. In 2018, the local population aged 20–79 years old from which we sampled was equal to 208,303. The mean age of this local population, according to the 2018 Central Statistical Office data, was 47.4 ± 15.8 years old, and 46.0% of the participants were male.

The percentages of the very high risk, high risk, moderate risk, and low CV risk classes were 17.6%, 13.5%, 22.8%, and 46.1%, respectively [8]. The mean Pol-SCORE risk was 3.98% (95% CI 3.54, 4.42), the mean FRS—Lipids was 8.42% (95% CI 7.82, 9.02), and the mean FRS–BMI was 10.73% (95% CI 10.03, 11.43). The mean LIFE-CVD 10-year risk and LIFE-CVD lifetime risk were 4.88% (95% CI 4.53, 5.24) and 17.31% (95% CI 16.56, 18.07), respectively. Detailed baseline data for the CV risk classes are presented in Table 2.

Using the optimal prevention model (Model 1), we can achieve a reduction of −1.74% (95% CI −1.46, −2.02) in the 10-year risk of CVD death (Pol-SCORE) in relation to the baseline model in the total population, a −0.85% (95% CI −0.72, −0.98) reduction approaching that of the moderate model of prevention (Model 2) and a −1.11% (95% CI −0.92, −1.30) reduction approaching that of the minimal model of prevention (Model 3). The 10-year risk of developing the first CVD event calculated with the FRS using the lipid model included a reduction of −3.28% (95% CI −2.97, −3.59) when using prevention model 1, −1.54% (95% CI −1.38, −1.70) when using prevention model 2, and −2.04% (95% CI −1.85, −2.23) when using prevention model 3 in the total population. Using the FRS based on BMI, we were able to predict risk reductions of −2.42% (95% CI −2.15, −2.69), −0.42% (95% CI −0.35, −0.49), and −1.74% (95% CI −1.54, −1.94) when using prevention models 1, 2, and 3, respectively. Similar results were found in the moderate, high, and very high CV risk categories. However, in the low CV risk class, the best model was the minimal model (risk reduction of −0.72% (95% CI −0.60, −0.84)), which showed even better performance than the optimal one (reduction of −0.69% (95% CI −0.55, −0.83)) using the FRS–BMI. Detailed data for the CV risk classes are presented in Table 3. Using the LIFE-CVD model in the optimal prevention strategy, the probability of survival-free heart attack or stroke increased by 2.01 (95% CI 1.85, 2.17) life years, and the reduction in the lifetime risk of MI, stroke, or CV death in the total population was −7.12% (95% CI −6.47, −7.78). In the moderate prevention strategy, the probability of survival-free of heart attack or stroke increased by 0.83 (95% CI 0.77, 0.90) life years; there was a −3.75% (95% CI −3.46, −4.05) reduction in the lifetime risk of MI, stroke, or CV death in the total population. In the minimal prevention strategy, the probability of survival free of heart attack or stroke increased by 1.21 (95% CI 1.10, 1.32) life years; there was a −3.49% (95% CI −3.29, −3.69) reduction in the lifetime risk of MI, stroke, or CV death in the total population. What is particularly salient in this analysis is a higher probability of survival years free of heart attack or stroke in the low CV risk category with the use of the minimal strategy (1.06 (95% CI 0.80, 1.32) life years) compared to the moderate (0.59 (95% CI 0.47, 0.72) life year) and even the optimal strategy (0.97 (95% CI 0.68, 1.27) life years) (Figure 2 Panel D). However, when using the minimal strategy, we were able to achieve a long-term CV risk reduction in LIFE-CVD of −2.21% (95% CI −1.96, −2.45); using the moderate models, we were able to achieve a −1.63% (95% CI −1.31, −1.95) reduction, and when using the optimal model, a −1.76% (95% CI −1.34, −2.17) reduction was able to be achieved. We obtained similar results by lowering the LIFE-CVD 10-year risk and LIFE-CVD lifetime risk of MI, stroke, and CV death in the low CV class. Using the optimal model of prevention, the calculated risk decreased by −0.23% (95% CI −0.17, −0.30); using the moderate model, the CV risk decreased by −0.20% (95% CI −0.16, −0.25); and using the minimum model, the CV risk decreased by −0.29% (95% CI −0.24, −0.35) for the 10-year risk of MI, stroke, and CV death. Detailed data for the particular CV risk classes are presented in Table 2 and Table 4 and in Figure 3.

In Table 5, we present the estimated number of the local inhabitants who would survive (Pol-SCORE) or who would not develop their first CVD event within 10 years (FRS) and would not develop MI, stroke, or CV death (LIFE-CVD) within 10 years or within their lifetime after applying the appropriate prevention strategies. Finally, in Table 6, we present the estimated CVD burden in the local population based on an earlier analysis [8]. We performed this analysis in order to be able to present these data to representatives of the authorities and the community more vividly. We want to emphasize the number of local people with improperly treated or unrecognized CV risk factors who, without the proper preventive programs, will develop complications or die from CVD.

## 4. Discussion

CVD is still a major cause of death and disability in European countries [1], including in Poland [14,15]. CVD is often silent and may occur suddenly [16,17], underscoring the importance of primary prevention. Estimating risk reduction using prevention strategies of various levels of intensity can help local authorities select or design prevention programs. Our findings highlight applying the minimal strategy to the overall population without previously diagnosed CVD is a more effective primarily prevention strategy than the application of the moderate strategy to the population with increased risk factors. Moreover, applying the minimal strategy to the low CV risk category would be more beneficial than following the optimal model.

### 4.1. Prevention Strategies

We have designed three different prevention models. The optimal model assumed the treatment thresholds for risk factors according to the European guidelines [2,18]. However, it is known that only a small number of the population is able to achieve the optimal therapy and target risk factor values. In the Polish population, only 10.9% of patients with hypercholesterolemia have achieved a total cholesterol level below 4.9 mmol/L, and only 5.4% have controlled comorbid hypertension and hypercholesterolemia [19]. Therefore, we designed other models. The moderate model assumed lowered BPs in people with hypertension, lowering the cholesterol level in people with hypercholesterolemia using moderate doses of the usual single drug [20,21], and eliminating obesity.

In Poland, over 40% of people with AH and 60% of people with hypercholesterolemia are not aware of their condition [22]. The minimal intervention model, which is based on lifestyle modification regardless of the baseline parameters, was designed for the entire analyzed population according to the criteria of each of the CV scales mentioned. In accordance with earlier surveys, a sodium reduction of about 1.75 g per day was associated with a reduction of 4.2/2.1 mmHg in BPs/BPd [23]. In obese people, every 10 kg reduction in body weight reduces LDL-C by 8 mg/dL [24,25]. Additionally, the Mediterranean diet significantly reduced BP and lipid levels. Adjusted changes from baseline in BPs were −2.3 (95% CI −4.0, −0.5) mmHg and −2.6 (95% CI −4.3, −0.9) mmHg in the Mediterranean diet with olive oil and the Mediterranean diet with nuts, respectively. Mean changes from baseline TC were −11.3, −13.6, and −4.4 mg/dL in the Mediterranean diet with olive oil and nuts, respectively [26]. Exercise has significant average effects on BPs and BPd [27,28]: aerobic training reduced BPs by –4 (95% CI, –5 to –2) mmHg and BPd by –3 (95% CI, –3 to –2) mmHg; resistance training reduced BPs by –2 (95% CI, –4 to 0) mmHg and BPd by –3 (95% CI, –5 to –2) mmHg, and combined training reduced BP by −3 (95% CI, −4 to −2) mmHg and BPd by −3 (95% CI, −3 to 0) mmHg. The effects of exercise on cholesterol have also been studied. Resistance training significantly reduced LDL-c by 6 (95% CI, −11 to −1) mg/dL [29]. On this basis, the minimal prevention strategy in the current study assumed a reduction in TC and LDL-C by 10%, BPs by 5 mmHg, BMI by 5%, and smoking cessation in all participants.

### 4.2. Estimating the Effects of Preventative Models

The prevention of primary CVD remains a significant challenge. By using the optimal prevention strategy, we should be able to obtain the best possible results for reducing CV risk. However, it is known that only a small number of the population achieves the optimal therapy and target risk factor values. In the EUROASPIRE V survey of primary prevention in Europe, risk factor control was poor in 43.5% of the patients with obesity; a total of 47% of patients on BP-lowering medication and 46.9% of dyslipidemic patients who were being treated for their condition reached their targets [4]. Patients with increased CV risk factors should be actively searched for and should have access to programs delivered by healthcare professionals, such as physicians, nurses, physiotherapists, dieticians, or physical activity specialists, that address all aspects of lifestyle (BP, lipids, glucose, physical activity, and adherence to medications) to reduce the risk of CV events. Despite the high involvement of healthcare professionals, risk reduction would still be lower than it would be in the proposed minimal intervention model. There is still a significant difference between evidence-based guidelines and daily clinical practice.

Our findings highlight that the minimal strategy applied to the overall population without CVD could be more effective than the moderate (realistic) strategy applied in the population with abnormal test results regardless of their CV risk category; even in the low CV risk category, the minimal intervention would be more beneficial than it would be the optimal one. It is worth emphasizing that almost half of the population belongs to the low-risk category. In general, CVD events occur after middle age, while most young adults have a low 10-year CVD risk. However, the early prevention of risk-modifying strategies can result in greater lifetime therapy benefits [30]. The LIFE-CVD [12] model provides a greater time-horizon than 10 years; hence, we could compare the effects of the assumed models not only in a 10-year interval but at a longer interval. Furthermore, lifetime estimates could be more comprehensive and approachable for patients or health providers. While using this model in the low CV risk category, non-pharmacological methods reduced the incidence of CVD more than the proposed optimal model in addition to the moderate preventative model. This study helped us to gain insights into the individual effects of prevention strategies and suggests that more attention should be paid to primary prevention for the entire population, which was also confirmed by Ma [31]. Environmental approaches involve the use of policy and structural changes to create environments where health is promoted and healthy choices are reinforced. Changes that make healthy behaviors easier and more convenient for individuals should be made to social and physical environments while maintaining broad reach and sustaining health benefits for the overall population. Environmental strategies that can help to reduce the CV risk factors include creating smoke-free environments and increasing access to healthier foods, including those with less sodium.

### 4.3. Strengths and Limitations

This study is limited by a sample from one region, which is an urban environment that may not be fully representative. For the calculation of the effects of particular strategies, we used scores used in primary prevention, whereas very high-risk individuals might have benefits similar to those seen in secondary prevention. Therefore, the results might underestimate the CVD risk population while simultaneously underestimating the effects of the interventions. The LIFE-CVD model uses the parental history of MI prior to age 60. In our data, we obtained information about the parental history of MI prior to age 50. Despite these limitations, there are some advantages to the study. We used data collected from a random sample of local citizens. The variables were collected using standardized questionnaires, methods, and equipment. The CV risk classes were carefully evaluated according to the latest recommendations: 2019 ESC/EAS guidelines for the management of dyslipidemias: lipid modification to reduce cardiovascular risk [2].

## 5. Conclusions

The percentages of the low, moderate, high, and very high CV risk classes in the general population are 46.1%, 22.8%, 13.5%, and 17.6%, respectively. A minimum strategy applied to the entire population without CVD may be more effective than a strategy that focuses on the moderate pharmacological treatment of individual risk factors, regardless of the initial CV risk category. In the low CV risk category, minimal (non-pharmacological lifestyle) intervention may be more favorable than optimal intervention. This suggests that the application of general principles of primary prevention, i.e., weight loss, diet modification, smoking cessation, and exercise, could produce better results than restricting treatment to patients with high BP and cholesterol levels. CV prevention requires modern prophylaxis programs that are adapted to medical and cultural conditions. Improvements in these areas can help create an environment in which people can receive high-quality care, make healthier choices, and take control of their health. These data could be relevant to public health institutions designing CV prevention strategies.

## Figures and Tables

**Figure 1 jcm-11-00688-f001:**
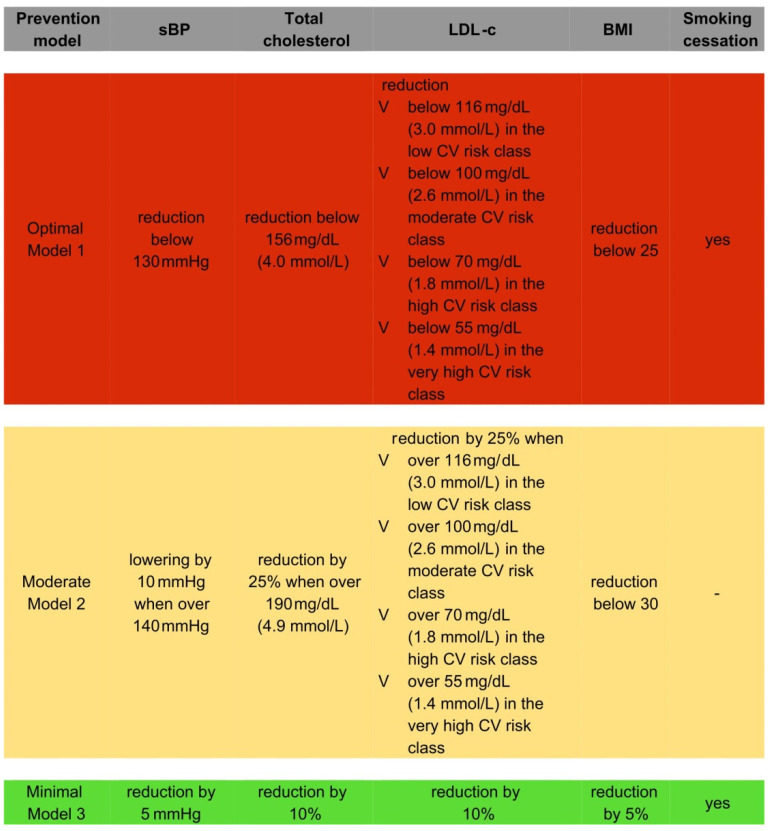
Prevention models designed.

**Figure 2 jcm-11-00688-f002:**
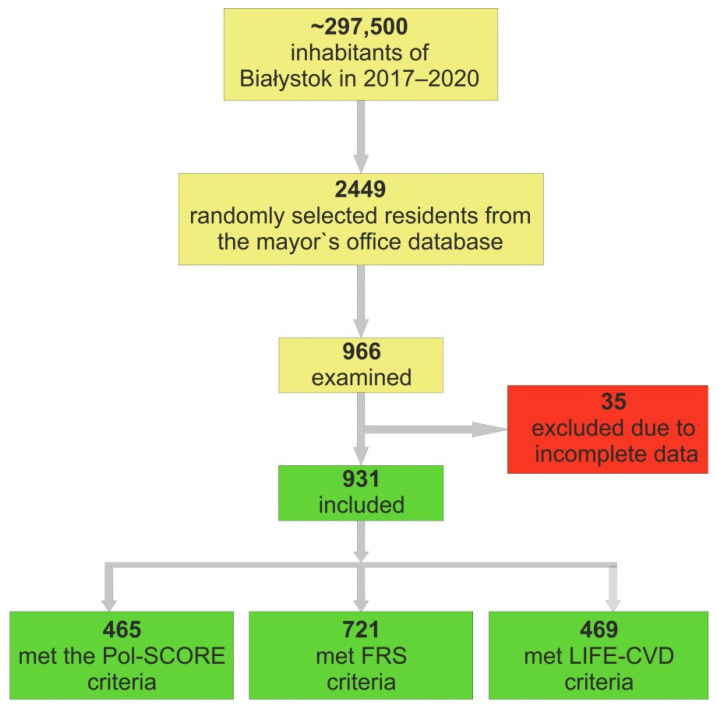
Analyzed cohort of included individuals.

**Figure 3 jcm-11-00688-f003:**
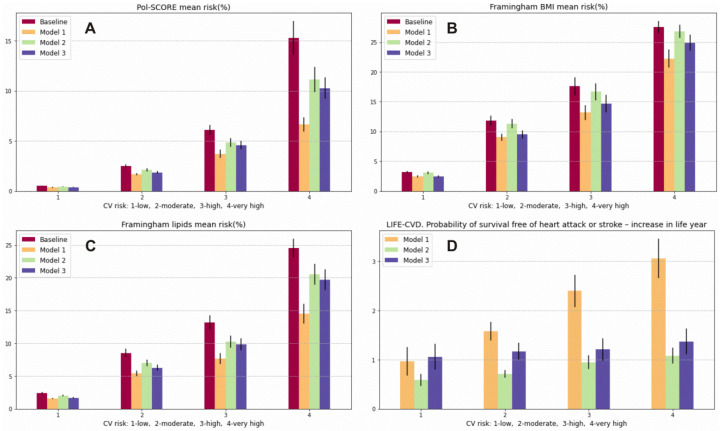
Prediction of 10-year risk of fatal CVD (Pol-SCORE): (**A**), prediction of 10-year risk of developing a first CVD event (Framingham Cardiovascular Disease) using the lipid model (**B**) and the BMI model; (**C**), probability of survival-free heart attack or stroke—increase in life years (LIFE-CVD) (**D**). Variables are presented as means and 95% confidence interval (95% CI).

**Table 1 jcm-11-00688-t001:** The baseline characteristics of the analyzed population.

Variable	Total Population *n* = 931
Age, years	49.1 ± 15.5
Male sex, *n*	402 (43.2)
BPs, mmHg	124.4 ± 17.7
BPd, mmHg	81.7 ± 10.1
BP ≥ 140 and/or ≥90 mmHg	253 (27.2)
HR, bpm	72.3 ± 10.9
Fasting glucose, mg/dL	102.1 ± 21.0
OGTT 120 min glucose, mg/dL	124.3 ± 39.7
HbA1c, %	5.5 ± 0.7
TC, mg/dL	192.5 ± 40.8
LDL-C, mg/dL	124.4 ± 37.8
HDL-C, mg/dL	62.6 ± 17.3
TG, mg/dL	113.2 ± 77.6
hs-CRP, mg/l	1.7 ± 4.2
Creatinine, μmol/L	70.9 ± 14.9
CrCl, mL/min	115.0 ± 40.7
LVEF Biplane, %	58.5 ± 5.7
BMI, kg/m^2^	26.8 ± 5.0
BMI < 25 kg/m^2^	330 (35.4)
BMI 25–29.99 kg/m^2^	352 (37.8)
BMI ≥ 30 kg/m^2^	249 (26.7)
History of hypertension	275 (29.6)
Undiagnosed hypertension *	107 (11.5)
History of hypercholesterolemia	290 (31.1)
Undiagnosed hypercholesterolemia **	399 (42.9)
History of diabetes	71 (7.6)
Undiagnosed diabetes ***	57 (6.1)
Currently smoking	186 (20.1)

Data are shown as *n* (%) and mean ± SD. BP: blood pressure; BPd: diastolic blood pressure; BPs: systolic blood pressure; bpm: beats per minute; CrCl: creatinine clearance using Cockcroft–Gault equation; HbA1c: hemoglobin A1c; HDL-C: high-density lipoprotein; HR: heart rate; hs-CRP; high-sensitivity C-reactive protein; kg: kilogram; LDL-C: low-density lipoprotein; mmHg, millimeters of mercury; LVEF Biplane: left ventricular ejection fraction biplane Simpson’s method; OGTT: oral glucose tolerance test; SD: standard deviation; TC: total cholesterol; TG: triglycerides. * BPs ≥ 140 and/or BPd ≥ 90 mmHg. ** TC > 190 mg/dL or LDL-c > 116 mg/dL in low CV class, >100 mg/dL in moderate CV class, >70 mg/dL in high CV class, and >55 mg/dL in very high CV class. *** Fasting glucose ≥ 126 mg/dL or OGGT 120 min glucose ≥ 200 mg/dL.

**Table 2 jcm-11-00688-t002:** Prediction of 10-year risk of fatal CVD (Pol-SCORE), 10-year risk of developing first CVD event (Framingham Cardiovascular Disease), and 10-year risk or life risk of MI, stroke, or CV death (lifetime-perspective model for individualizing cardiovascular disease prevention strategies in apparently healthy people: LIFE-CVD).

Prevention Strategy	Scales	CV Risk Classes	*n*	Average Risk (%) Mean (95% CI)
Baseline	Pol-SCORE	Low	126	0.51 (0.47–0.55)
		Moderate	201	2.50 (2.35–2.65)
		High	90	6.10 (5.62–6.58)
		Very high	48	15.29 (13.60–16.98)
		Total	465	3.98 (3.54–4.42)
	FRS—Lipids	Low	319	2.40 (2.22–2.58)
		Moderate	205	8.55 (7.94–9.16)
		High	113	13.18 (12.08–14.28)
		Very high	84	24.53 (23.10–25.96)
		Total	721	8.42 (7.82–9.02)
	FRS—BMI	Low	319	3.15 (2.91–3.39)
		Moderate	205	11.84 (11.02–12.66)
		High	113	17.59 (16.06–19.12)
		Very high	84	27.54 (26.55–28.53)
		Total	721	10.73 (10.03–11.43)
	LIFE-CVD 10-year risk	Low	70	1.32 (1.16–1.48)
		Moderate	182	3.24 (3.04–3.44)
		High	114	4.98 (4.68–5.29)
		Very high	103	10.09 (9.17–11.01)
		Total	469	4.88 (4.53–5.24)
	LIFE-CVD lifetime risk	Low	70	11.46 (10.66–12.27)
		Moderate	182	16.42 (15.45–17.39)
		High	114	17.76 (16.22–19.30)
		Very high	103	22.37 (20.40–24.34)
		Total	469	17.31 (16.56–18.07)

The data are shown as the means and 95% confidence interval (95% CI). BMI, body mass index; CV, cardiovascular; CVD, cardiovascular disease FRS, Cardiovascular Disease Framingham Heart Study; LIFE-CVD, lifetime-perspective model for individualizing cardiovascular disease prevention strategies in apparently healthy people; SCORE, Systematic Coronary Risk Estimation.

**Table 3 jcm-11-00688-t003:** Estimation of the effects of particular prevention models on 10-year risk of fatal CVD (Pol-SCORE); 10-year risk of developing the first CVD event (Framingham Cardiovascular Disease); and 10-year risk or life risk of MI, stroke, or CV death (lifetime-perspective model for individualizing cardiovascular disease prevention strategies in apparently healthy people LIFE-CVD) in the appropriate study population.

Prevention Strategy	Scales	CV Risk Classes	*n*	Average Risk (%) Mean (95% CI)	The Absolute Value of the Reduction from Baseline Risk (%) Mean (95% CI)
Model 1 Optimal	Pol-SCORE	Low	126	0.37 (0.34–0.40)	−0.14 (0.12–0.16)
		Moderate	201	1.69 (1.57–1.81)	−0.81 (0.71–0.91)
		High	90	3.71 (3.30–4.12)	−2.40 (2.07–2.73)
		Very high	48	6.66 (5.94–7.38)	−8.63 (7.16–10.10)
		Total	465	2.24 (2.03–2.45)	−1.74 (1.46–2.02)
	FRS—Lipids	Low	319	1.59 (1.49–1.69)	−0.82 (0.71–0.93)
		Moderate	205	5.42 (5.03–5.81)	−3.13 (2.76–3.50)
		High	113	7.67 (6.86–8.48)	−5.51 (4.81–6.21)
		Very high	84	14.55 (13.03–16.07)	−9.98 (8.73–11.23)
		Total	721	5.14 (4.75–5.53)	−3.28 (2.97–3.59)
	FRS—BMI	Low	319	2.47 (2.32–2.62)	−0.69 (0.55–0.83)
		Moderate	205	9.03 (8.43–9.63)	−2.81 (2.33–3.29)
		High	113	13.17 (11.90–14.44)	−4.43 (3.64–5.22)
		Very high	84	22.23 (20.72–23.74)	−5.31 (4.15–6.47)
		Total	721	8.31 (7.74–8.88)	−2.42 (2.15–2.69)
	LIFE-CVD 10-year risk	Low	70	1.08 (0.95–1.21)	−0.23 (0.17–0.30)
		Moderate	182	2.28 (2.13–2.43)	−0.96 (0.85–1.07)
		High	114	2.69 (2.45–2.93)	−2.29 (2.05–2.53)
		Very high	103	3.77 (3.43–4.10)	−6.32 (5.49–7.15)
		Total	469	2.53 (2.39–2.66)	−2.35 (2.07–2.63)
	LIFE-CVD lifetime risk	Low	70	9.71 (9.06–10.35)	−1.76 (1.34–2.17)
		Moderate	182	11.82 (11.18–12.46)	−4.60 (4.04–5.16)
		High	114	9.25 (8.66–9.84)	−8.51 (7.25–9.76)
		Very-high	103	8.67 (8.03–9.32)	−13.70 (11.93–15.46)
		Total	469	10.19 (9.83–10.54)	−7.12 (6.47–7.78)
Model 2 Moderate	Pol-SCORE	Low	126	0.44 (0.40–0.48)	−0.07 (0.06–0.08)
		Moderate	201	2.12 (1.98–2.26)	−0.38 (0.33–0.43)
		High	90	4.85 (4.42–5.28)	−1.26 (1.07–1.45)
		Very high	48	11.16 (9.91–12.41)	−4.13 (3.46–4.80)
		Total	465	3.13 (2.80–3.46)	−0.85 (0.72–0.98)
	FRS—Lipids	Low	319	2.01 (1.87–2.15)	−0.40 (0.34–0.46)
		Moderate	205	6.98 (6.46–7.50)	−1.57 (1.34–1.80)
		High	113	10.24 (9.33–11.15)	−2.94 (2.52–3.36)
		Very high	84	20.57 (18.99–22.15)	−3.96 (3.27–4.65)
		Total	721	6.88 (6.37–7.39)	−1.54 (1.38–1.70)
	FRS—BMI	Low	319	3.07 (2.85–3.29)	−0.08 (0.05–0.11)
		Moderate	205	11.30 (10.53–12.07)	−0.54 (0.38–0.70)
		High	113	16.66 (15.21–18.11)	−0.93 (0.68–1.18)
		Very high	84	26.83 (25.74–27.92)	−0.71 (0.40–1.02)
		Total	721	10.31 (9.64–10.98)	−0.42 (0.35–0.49)
	LIFE-CVD 10-year risk	Low	70	1.11 (0.99–1.24)	−0.20 (0.16–0.25)
		Moderate	182	2.66 (2.48–2.83)	−0.58 (0.52–0.65)
		High	114	3.82 (3.56–4.08)	−1.16 (1.05–1.28)
		Very high	103	7.18 (6.57–7.79)	−2.91 (2.54–3.27)
		Total	469	3.70 (3.46–3.95)	−1.18 (1.05–1.30)
	LIFE-CVD lifetime risk	Low	70	9.83 (9.21–10.45)	−1.63 (1.31–1.95)
		Moderate	182	13.50 (12.77–14.23)	−2.92 (2.59–3.26)
		High	114	13.48 (12.51–14.46)	−4.27 (3.64–4.90)
		Very high	103	16.28 (14.98–17.58)	−6.09 (5.32–6.86)
		Total	469	13.56 (13.05–14.06)	−3.75 (3.46–4.05)
Model 3 Minimal	Pol-SCORE	Low	126	0.38 (0.35–0.41)	−0.13 (0.11–0.15)
		Moderate	201	1.87 (1.74–2.00)	−0.63 (0.55–0.71)
		High	90	4.60 (4.17–5.03)	−1.50 (1.25–1.75)
		Very high	48	10.28 (9.20–11.36)	−5.01 (3.71–6.31)
		Total	465	2.87 (2.57–3.17)	−1.11 (0.92–1.30)
	FRS—Lipids	Low	319	1.69 (1.56–1.82)	−0.71 (0.62–0.80)
		Moderate	205	6.29 (5.82–6.76)	−2.26 (1.98–2.54)
		High	113	9.84 (8.91–10.77)	−3.34 (2.86–3.82)
		Very high	84	19.71 (18.09–21.33)	−4.82 (3.83–5.81)
		Total	721	6.38 (5.88–6.88)	−2.04 (1.85–2.23)
	FRS—BMI	Low	319	2.43 (2.26–2.60)	−0.72 (0.60–0.84)
		Moderate	205	9.51 (8.81–10.21)	−2.33 (1.93–2.73)
		High	113	14.70 (13.23–16.17)	−2.89 (2.25–3.53)
		Very high	84	24.89 (23.54–26.24)	−2.64 (1.74–3.54)
		Total	721	8.98 (8.34–9.62)	−1.74 (1.54–1.94)
	LIFE-CVD 10-year risk	Low	70	1.03 (0.90–1.15)	−0.29 (0.24–0.35)
		Moderate	182	2.48 (2.32–2.64)	−0.76 (0.68–0.85)
		High	114	3.81 (3.56–4.06)	−1.17 (1.02–1.33)
		Very high	103	7.64 (6.82–8.46)	−2.45 (2.15–2.75)
		Total	469	3.71 (3.42–3.99)	−1.16 (1.05–1.26)
	LIFE-CVD lifetime risk	Low	70	9.26 (8.54–9.97)	−2.21 (1.96–2.45)
		Moderate	182	13.19 (12.33–14.06)	−3.23 (2.98–3.47)
		High	114	14.12 (12.81–15.43)	−3.64 (3.20–4.07)
		Very high	103	17.65 (15.91–19.40)	−4.72 (4.17–5.26)
		Total	469	13.79 (13.14–14.44)	−3.49 (3.29–3.69)

The data are shown as means and 95% confidence intervals (95% CI). MI, myocardial infarction; BMI, body mass index; BPs, systolic blood pressure; CV, cardiovascular; CVD, cardiovascular disease; FRS, Cardiovascular Disease Framingham Heart Study; LIFE-CVD, lifetime-perspective model for individualizing cardiovascular disease prevention strategies in apparently healthy people; SCORE, Systematic Coronary Risk Estimation; TC, total cholesterol. Model 1: optimal prevention strategy: BPs < 130 mmHg, TC < 156 mg/dL, BMI < 25, smoking cessation; Model 2: moderate prevention strategy: BPs lowering by 10 mmHg when >140 mmHg, TC lowering by 25% when >190 mg/dL, BMI < 30. Model 3: minimal prevention strategy: TC lowering by 10%, BPs lowering by 5 mmHg, BMI lowering by 5%, smoking cessation.

**Table 4 jcm-11-00688-t004:** Prediction of lifetime benefit using the lifetime-perspective model for individualizing cardiovascular disease prevention strategies in apparently healthy people (LIFE-CVD) in the appropriate study population.

Prevention Strategy	CV Risk Classes	*n*	Probability of Survival Free of Heart Attack or Stroke—Increase in Life Years in Relation to the Baseline Mean (95% CI)
Model 1 Optimal	Low	70	0.97 (0.68–1.27)
	Moderate	182	1.58 (1.38–1.77)
	High	114	2.40 (2.06–2.73)
	Very high	103	3.06 (2.66–3.46)
	Total	469	2.01 (1.85–2.17)
Model 2 Moderate	Low	70	0.59 (0.47–0.72)
	Moderate	182	0.71 (0.63–0.79)
	High	114	0.95 (0.81–1.10)
	Very high	103	1.08 (0.92–1.25)
	Total	469	0.83 (0.77–0.90)
Model 3 Minimal	Low	70	1.06 (0.80–1.32)
	Moderate	182	1.17 (1.01–1.34)
	High	114	1.21 (0.98–1.45)
	Very high	103	1.37 (1.12–1.63)
	Total	469	1.21 (1.10–1.32)

The data are shown as means and 95% confidence intervals (95% CI). MI, myocardial infarction; CV, cardiovascular; BPs, systolic blood pressure; TC, total cholesterol; LDL-C, low-density lipoprotein cholesterol. Model 1: optimal prevention strategy: BPs < 130 mmHg, LDL-C in low CV risk class < 116 mg/dL, in moderate CV risk class < 100 mg/dL, in high CV risk class < 70 mg/dL, in very high CV risk class < 55 mg/dL, smoking cessation; Model 2: moderate prevention strategy: BPs lowering by 10 mmHg when >140 mmHg, LDL-C lowering by 25% when low CV risk class > 116 mg/dL, in moderate CV risk class > 100 mg/dL, in high CV risk class > 70 mg/dL, in very high CV risk class > 55 mg/dL; Model 3: minimal prevention strategy: TC lowering by 10%, LDL-C lowering by 10%, BPs lowering by 5 mmHg, smoking cessation.

**Table 5 jcm-11-00688-t005:** Estimated number of local the inhabitants who would survive (Pol-SCORE) or who would not develop a first CVD event within 10 years (Cardiovascular Disease Framingham Heart Study) or who would not develop MI, stroke, or CV death (lifetime-perspective model for individualizing cardiovascular disease prevention strategies in apparently healthy people LIFE-CVD) within 10 years or within their lifetime after applying prevention strategies.

SCORES		Baseline	Model 1	Model 2	Model 3
Pol-SCORE	Deceased	3963	2177	3095	2815
	Survivors	-	1786	868	1148
FRS—Lipids	First CVD event within 10 years	13,148	7987	10,734	9880
	Stay free of developing the first CVD event within 10 years	-	5161	2414	3268
FRS—BMI	First CVD event within 10 years	16,727	12,835	16,069	13,927
	Stay free of developing the first CVD event within 10 years	-	3892	658	2800
LIFE-CVD 10-year risk	First MI, stroke, or CV death within 10 years	4934	2534	3738	3748
	Stay free of developing the first CVD event within 10 years	-	2400	1196	1186
LIFE-CVD lifetime risk	First MI, stroke, or CV death within lifetime	17,461	10,186	13,640	13,899
	Stay free of developing the first CVD event within lifetime	-	7275	3821	3562

BMI, body mass index; BPs, systolic blood pressure; CV, cardiovascular; CVD, cardiovascular disease, TC, total cholesterol; FRS, Cardiovascular Disease Framingham Heart Study; SCORE Systematic Coronary Risk Estimation. Model 1: optimal prevention strategy: BPs < 130 mmHg, TC < 156 mg/dL, BMI < 25, smoking cessation; Model 2: moderate prevention strategy: BPs lowering by 10 mmHg when >140 mmHg, TC lowering by 25% when >190 mg/dL, BMI < 30. Model 3: minimal prevention strategy: TC lowering by 10%, BPs lowering by 5 mmHg, BMI lowering by 5%, smoking cessation.

**Table 6 jcm-11-00688-t006:** Estimated cardiovascular disease burden in the local population.

Variable	Total Population	Cardiovascular Risk Class			
		Low	Moderate	High	Very High
Local population	204,511	98,378	44,434	26,476	35,223
History of hypertension	58,283	7173	15,995	12,394	22,721
Uncontrolled BP in patients diagnosed with hypertension *	41,934	4767	11,286	8644	17,237
Undiagnosed hypertension	77,746	40,470	19,782	8409	9085
History of hypercholesterolemia	60,829	12,521	20,005	11,669	16,634
Uncontrolled lipid profile in patients with diagnosed hypercholesterolemia **	53,370	10,504	15,674	10,996	16,196
Undiagnosed hypercholesterolemia ***	91,293	36,790	22,168	14,175	18,160
History of diabetes	15,220	512	3720	2930	8058
Uncontrolled glucose in patients diagnosed with diabetes ****	4493	256	457	1119	2661
Undiagnosed diabetes *****	522	0	257	265	0

BP: blood pressure; BPs: systolic blood pressure; BPd: diastolic blood pressure; CV: cardiovascular; HbA1c: hemoglobin A1c; LDL: low-density lipoprotein; TC: total cholesterol. * BPs < 130 and BPd < 80 mmHg below 65 years old, BPs < 140 and BPd < 80 mmHg 65–80 years old, BPs < 150 and BPd < 80 mmHg over 80 years old. ** LDL-c < 116 mg/dL in low CV class, <100 mg/dL in moderate CV class, <70 mg/dL in high CV class, <55 mg/dL in very high CV class. *** TC > 190 mg/dL or LDL-c > 116 mg/dL in low CV class, >100 mg/dL in moderate CV class, >70 mg/dL in high CV class, >55 mg/dL in very high CV class. **** HbA1c <7.0%. ***** Fasting glucose ≥ 126 mg/dL or OGGT 120 min glucose ≥ 200 mg/dL.

## Data Availability

The data set we generated during and/or analyzed during the current study are not publicly available due to confidentiality issues but are available from the corresponding author on request.

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
