# Peer review of "Effectiveness of Lifestyle Modification vs. Therapeutic, Preventative Strategies for Reducing Cardiovascular Risk in Primary Prevention—A Cohort Study"

_jcm, 2022, doi:10.3390/jcm11030688_

Round 1
Reviewer 1 Report
Thank you for your careful consideration of reviews' comments. I think the authors have properly addressed most of my previous comments. However, minor revision is still needed.
1. In the abstract, the authors should present how much reduction was achieved.
2. The authors have neglected my previous comment about the study objective. Please specify the objective of this study at the end of the introduction section.
Author Response
Comments to Author:
Reviewer #1
Thank you for your careful consideration of reviews' comments. I think the authors have properly addressed most of my previous comments. However, minor revision is still needed.
- In the abstract, the authors should present how much reduction was achieved.
The following has been added to the abstract section: ”e.g. using the optimal model of prevention (Model 1), in the total population, we can achieve a reduction of -1.74% of the 10-year risk of CVD death (SCORE) in relation to the baseline model, a -0.85% reduction approaching the moderate model of prevention (Model 2), and a -1.11% reduction approaching the minimal model of prevention (Model 3). However, in the low CV risk class, the best model was the minimal one (risk reduction of -0.72%) even better than the optimal one (reduction of - 0.69%) using the FRS.”
- The authors have neglected my previous comment about the study objective. Please specify the objective of this study at the end of the introduction section.
The following has been added to the introduction section: “The aim of this study was to calculate the potential for CV risk reduction using three different prevention strategies outlined above to evaluate the effect of primary prevention using validated and well-known scales across a total population as well as by CV risk categories.”
All changes to the manuscript are marked in red.
Thank you for your review, I am sure it improves the quality of the manuscript.
Kind regards,
Karol Kaminski
Reviewer 2 Report
Dear Authors, I appreciate your efforts and, as far as I am concerned, the paper has been improved. However, the English style remains too poor, and it needs an extensive editing. A native speaker revision should be necessary. I have highlighted only a few examples

Author Response
Comments to Author:
Reviewer #2
Dear Authors, I appreciate your efforts and, as far as I am concerned, the paper has been improved. However, the English style remains too poor, and it needs an extensive editing. A native speaker revision should be necessary. I have highlighted only a few examples.
Dear Reviewer, thank you for your valuable suggestions. The linguistic revision of this manuscript was performed twice. Current version has undergone English language editing by MDPI. The text has been checked for correct use of grammar and common technical terms, and edited to a level suitable for reporting research in a scholarly journal. MDPI uses experienced, native English speaking editor. We attach a certificate. If this is not enough, we will try to improve the English language again.
All changes to the manuscript are marked in red.
Thank you for your review, I am sure it improves the quality of the manuscript.
Kind regards,
Karol Kaminski

Reviewer 3 Report
This is a resubmitted manuscript. The authors did not provide responses to the comments from previous reviewers. Therefore, it is not possible to evaluate the improvement of the manuscript as compared to the original version. Nonetheless, I have some concerns regarding the study:
- In general, both the abstract and the content are very confusing and exhausting. I read them back and forth multiple times and still don't understand what the authors tried to do in this study. Why did they used 3 different scoring system and how they classified those participants into 3 groups (i.e., preventive strategies). I think the best way to deal with this is to ask for someone else who was not involved in this study to read the draft before submission and see if this person could actually understand this study.
- Line 18: the aim was unclear. "The aim of this study was to calculate the potential for CV risk reduction when using three different prevention strategies to evaluate the effect of primary prevention." what are those three strategies?
- It is bizzare that the title was not consistent with the aim of the study. The title specifically said that the authors compared lifestyle modification vs. therapeutic, preventative strategies. This is very confusing. Please clarify.
- Line 20: 20-79 years? or months?
- Line 25: "190mg%" I never see such a unit.
- "The optimal prevention strategy assumed the attainment of therapeutic goals according to the European guidelines.", which guideline?
- "The minimal prevention strategy assumed lowering the CV risk by lifestyle modification" and who were the target participants? also people with increased risk factors as used in the moderate strategy?
- "Moreover, applying the minimal strategy to the low CV risk class population was similarly beneficial to following even the optimal model" Why is this even surprising?
- "A strategy based on lifestyle modification in a population without established CVD could be more effective than the moderate strategy." Is the comparison between those 3 strategies even apple-to-apple? I am confused.
- Lines 66-76: What are the basis of those cut-off values? Who defined those? Which guidelines? Who declared those cut-off as realistic, non-realistic, idealistic, attainable or not?
- "In the optimal (therapeutic prevention strategy (Model 1)), we ideally assumed attainment of therapeutic goals in patients with present CV risk factors" this is not entirely true because even in the moderate and minimal groups, the authors also included people with CV risk.
- "Calculating risk reduction by using different degrees of prevention intensity can help local authorities to improve their prevention programs. The use of real-time CV risk data is a powerful factor in the optimization of evidence-based policy. This data-driven approach can enable responsible people to choose the right health strategies that can lead to better health for all." How? I think these are way too far from what the authors obtained in this study.
Overall, please improve the readability and clarity of this study. If the authors could not do it, ask a help from a preventive Cardiologist.
Also, I think it is better to focus on one aspect that needs to be highlighted. Having those data in a manuscript with poor clarity like this is not going anywhere.
Author Response
Comments to Author:
Reviewer #3
This is a resubmitted manuscript. The authors did not provide responses to the comments from previous reviewers. Therefore, it is not possible to evaluate the improvement of the manuscript as compared to the original version. Nonetheless, I have some concerns regarding the study:
- In general, both the abstract and the content are very confusing and exhausting. I read them back and forth multiple times and still don't understand what the authors tried to do in this study. Why did they used 3 different scoring system and how they classified those participants into 3 groups (i.e., preventive strategies). I think the best way to deal with this is to ask for someone else who was not involved in this study to read the draft before submission and see if this person could actually understand this study.
Dear Reviewer, we are sorry that work is confusing. In the new version we elaborated more on the aim of the study and corrected language. Additional information about our research, including the characteristics of the population regarding the division into different risk categories was described in detail in the previous work entitled "A Similar Lifetime CV Risk and a Similar Cardiometabolic Profile in the Moderate and High Cardiovascular Risk Populations: A Population-Based Study" doi.org/10.3390/jcm10081584. In this work we did not want to repeat the previous one.
Due to the fact that our cohort is examined in a very detailed way (including ultrasound of the carotid arteries, which is rare), it allows for a precise division into cardiovascular risk categories. We think it is very unique and valuable. The calculated primary cardiovascular risk allowed us to take further steps to use the scales already used to assess the primary baseline cardiovascular risk to estimate the possibility of risk reduction using the prevention models we proposed. A large part of the authors are actively practicing cardiologists who use cardiovascular risk calculators on a daily basis.
- Line 18: the aim was unclear. "The aim of this study was to calculate the potential for CV risk reduction when using three different prevention strategies to evaluate the effect of primary prevention." what are those three strategies?
Prevention strategies have been established by the authors. We described them in the introduction section: “We designed three different prevention models. In the optimal (therapeutic prevention strategy (Model 1)), we ideally assumed attainment of therapeutic goals in patients with present CV risk factors: successful BPs reduction below 130mmHg, LDL-C reduction below 116mg% in the low CV risk class, below 100mg% in the moderate CV risk class, below 70mg% in the high CV risk class, below 55mg% in the very high CV risk class, TC level reduction below 156mg%, BMI reduction below 25, and smoking cessation. In the moderate therapeutic prevention strategy (Model 2), we assumed more realistic therapeutic goals: successful BPs lowering by 10mmHg when over 140mmHg; LDL-C reduction by 25% when over 116mg% in the low CV risk class, over 100mg% in the moderate CV risk class, over 70mg% in the high CV risk class, over 55mg% in the very high CV risk class; TC level reduction by 25% when over 190mg%; and BMI reduction below 30. In the minimal prevention strategy (a population-based approach (Model 3)), we assumed risk factors modifications that are attainable by lifestyle modifications, namely lowering the following values in the analyzed population: TC and LDL-C levels by 10%, BPs by 5mmHg, and BMI by 5% accompanied by smoking cessation. “
- It is bizzare that the title was not consistent with the aim of the study. The title specifically said that the authors compared lifestyle modification vs. therapeutic, preventative strategies. This is very confusing. Please clarify.
The title refers to the result of the study. Since the first strategies were based on non-pharmacological and pharmacological management, the third and last strategy was solely lifestyle modification. The third strategy turned out to be better than the second in all risk scales. The second strategy is one that reflects everyday life, which we deal with in our professional work, and as it turns out not only in our study, it does not bring sufficient population benefits, whereas intensification of lifestyle modification based – addressing the whole population, might be more efficient.
- Line 20: 20-79 years? or months?
Has been corrected: “A total of 931 individuals aged 20-79 years from the Bialystok PLUS Study were analyzed.”
- Line 25: "190mg%" I never see such a unit.
In our labs, this non SI unit is more popular and it is the same as mg/dL. In the new version of the manuscript we changed into this unit. For better understanding, we also added mmol/L in the introductory section.
- "The optimal prevention strategy assumed the attainment of therapeutic goals according to the European guidelines.", which guideline?
In the discussion section, we described: “We have designed three different prevention models. The optimal model assumed the treatment thresholds for risk factors according to the European guidelines [2,18].”
2.Mach, F.; Baigent, C.; Catapano, A.L.; Koskinas, K.C.; Casula, M.; Badimon, L.; Chapman, M.J.; De Backer, G.G.; Delgado, V.; Ference, B.A., et al. 2019 ESC/EAS Guidelines for the management of dyslipidaemias: lipid modification to reduce cardiovascular risk. Eur Heart J 2020, 41, 111-188, doi:10.1093/eurheartj/ehz455.
18.Williams, B.; Mancia, G.; Spiering, W.; Agabiti Rosei, E.; Azizi, M.; Burnier, M.; Clement, D.L.; Coca, A.; de Simone, G.; Dominiczak, A., et al. 2018 ESC/ESH Guidelines for the management of arterial hypertension. Eur Heart J 2018, 39, 3021-3104, doi:10.1093/eurheartj/ehy339.
- "The minimal prevention strategy assumed lowering the CV risk by lifestyle modification" and who were the target participants? also people with increased risk factors as used in the moderate strategy?
As mentioned above, our real population has been divided precisely according to the “2019 ESC / EAS guidelines for the management of dyslipidemias: lipid modification to reduce cardiovascular risk”. Population with moderate CV risk according to these guidelines: "Young patients (T1DM <35 years; T2DM <50 years) with DM duration <10 years, without other risk factors. Calculated SCORE> _1% and <5% for 10-year risk of fatal CVD. "
- "Moreover, applying the minimal strategy to the low CV risk class population was similarly beneficial to following even the optimal model" Why is this even surprising?
This might be surprising for potential readers who are not experienced with cardiovascular prevention because for most categories, lifestyle modification and pharmacotherapy were required to reduce risk in this way. It is surprising that in this category, the use of lifestyle modification alone is sufficient to lower cardiovascular risk. This is a very important aspect for specialists involved in creating a prevention strategy, to place great emphasis on changing lifestyle, not on pharmacological treatment and overloading the medical care system. For epidemiologists and cardiologists with experience in cardiovascular prevention it might be less surprising.
- "A strategy based on lifestyle modification in a population without established CVD could be more effective than the moderate strategy." Is the comparison between those 3 strategies even apple-to-apple? I am confused.
A strategy based on a lifestyle change, as explained above, may be based on population rather than pharmacological treatment in people with elevated blood pressure or an abnormal lipid profile.
- Lines 66-76: What are the basis of those cut-off values? Who defined those? Which guidelines? Who declared those cut-off as realistic, non-realistic, idealistic, attainable or not?
As mentioned above, we have chosen the prevention strategies ourselves. We proposed Model 1 based on European guidelines, Model 2 based on medical practice and medical data, and Model 3 based on population data in current publications.
- "In the optimal (therapeutic prevention strategy (Model 1)), we ideally assumed attainment of therapeutic goals in patients with present CV risk factors" this is not entirely true because even in the moderate and minimal groups, the authors also included people with CV risk.
Yes, there are individuals with known cardiovascular disease in the high and very high cardiovascular risk group. But individuals with diagnosed cardiovascular diseases were not eligible for the final analysis. This is detailed in section Calculators for the assessment of CV risk in primary prevention:
“The Systematic Coronary Risk Estimation (SCORE) was calculated, excluding participants who were pre-qualified in the high and very high CV risk classes according to the aforementioned recommendations [2]. We used Pol-SCORE system, because it was recalibrated in Poland [9,10]. According to the original risk stratification in the ESC guidelines, participants with previously diagnosed CVD (myocardial infarction—MI, ischemic heart disease—IHD, stroke, transient ischemic attack—TIA, peripheral arterial disease—PAD, significant plaque on carotid ultrasound >50%), DM previously diagnosed or at the time of study entry, moderate or severe chronic kidney diseases (CKD) at the time of study entry, markedly elevated single risk factor (TC >310mg%, LDL-c >190mg%, BP ≥ 180/110mmHg) at the time of study entry, and younger than 40 years old or older than 70 years old were excluded from the Pol-SCORE calculation, as they present higher CV risk than that calculated according to the algorithm.
Cardiovascular Disease Framingham Heart Study (FRS) predicted a 10-year risk of developing the first CVD event (coronary heart disease, stroke, peripheral artery disease, or heart failure) using scores for BMI or lipids based on the following factors: age, smoking, diabetes, treated and untreated BPs, TC, high-density lipoprotein cholesterol (HDL-C), or lipids replacing BMI [11]. Participants with previously diagnosed CVD (myocardial infarction—MI, ischemic heart disease—IHD, stroke, transient ischemic attack—TIA, peripheral arterial disease—PAD, significant plaque on carotid ultrasound >50%), and younger than 30 years old or older than 74 years old were excluded from the further analysis.
To assess lifetime risk, the LIFE-CVD was used. This model calculates a 10-year risk of myocardial infarction (MI), stroke, or CV death; lifetime risk of MI, stroke, or CV death; or the probability of survival free of heart attack or stroke using the following factors: age, gender, smoking, geographic region, diabetes, parental history of MI prior to age 60, BPs, BMI, TC, HDL-C, and LDL-C [12]. Participants with previously diagnosed CVD (MI, IHD, stroke, TIA, peripheral arterial disease, significant plaque on carotid ultrasound >50%), and younger than 45 years old or older than 79 years old were excluded from this analysis.”
- "Calculating risk reduction by using different degrees of prevention intensity can help local authorities to improve their prevention programs. The use of real-time CV risk data is a powerful factor in the optimization of evidence-based policy. This data-driven approach can enable responsible people to choose the right health strategies that can lead to better health for all." How? I think these are way too far from what the authors obtained in this study.
In our opinion, the above analysis gives a great insight into the possibilities of reducing cardiovascular risk with different population strategies. Our work is based on a very thorough and detailed population assessment and the risk has been estimated using well-known and verified calculators.
All changes to the manuscript are marked in red.
Thank you for your review, I am sure it improves the quality of the manuscript.
Kind regards,
Karol Kaminski
Round 2
Reviewer 3 Report
Thank you for the explanation.
Author Response
We again thank the Reviewer for comments on our manuscript.
We are sure the comments have improved the quality of the manuscript.

This manuscript is a resubmission of an earlier submission. The following is a list of the peer review reports and author responses from that submission.
Round 1
Reviewer 1 Report
The study aimed to assess potential effectiveness of lifestyle modification versus therapeutic preventive strategies for reducing the risk of cardiovascular disease. This topic has potential implication for clinical practice and public health. However, the quality of the study requires substantial improvement. Below are my comments.
- Title
(1) It is unclear what the comparative strategies were for. Please specify “for reducing cardiovascular risk”.
(2) Please specify the study type.
- Abstract
(1) The results section was completely missing.
(2) In the methods section, the authors should have specified study design, outcomes and statistical methods
- Introduction
(1) The author should have specified the objectives of this study.
(2) If there have been relevant epidemiological studies on a similar topic, these studies should be presented in the introduction and the authors should have justified why to conduct the present study.
- Methods
(1) The number of local residents was 297 403, among whom 2449 randomly selected residents from the mayor’s office database were invited. Finally, 931 individuals were analyzed. It is unclear how the random selection was performed. Furthermore, it is unclear why only 931 of 2449 individuals contributed to the analysis. Selection bias may occur in this sample selection process.
(2) The authors designed three different prevention models: optimal, moderate and minimal. Please provide supporting evidence to justify this classification of prevention models.
(3) Was there any confounding adjusted in the statistical modelling? Please specify.
(4) To answer the research question, descriptive analysis was not enough. I would suggest adding Cox regression model.
(5) It seems that the authors simply exclude participants with missing data. However, they should have properly dealt with missing data.
- Results
(1) A flowchart of participant inclusion should be provided.
(2) A table to present the baseline characteristics should be provided.
(3) The main findings are mostly descriptive. To show which prevention strategy was more effective, the authors should estimate an effect size through direct comparison between three prevention strategies, e.g., risk ratio or risk difference.
- Discussion
(1) The section “4.1.Prevention strategies” presents many results of previous studies, which could be simplified.
(2) Could the authors add some implications for future research?
Reviewer 2 Report
The topic could be interesting but the study is strongly affected by significant drawbacks. The paper should be totally revised and rewritten. English style must be massively improved. Study design is poor with subheadings not well defined and overlapping. The discussion is poor and not focused.
Reviewer 3 Report
In this study, Dr. Chlabicz and colleagues tried to calculate the potential CV risk reduction using theoretical prevention strategies (based on abstract). Although this can be an interesting study to pursue, I am confused by many things in this paper and partly, it was because of the poorly written manuscript.
- Certainly, English needs to be improved. A check by professional English editing is suggested.
- What does it mean by "theoretical prevention strategies"?
- The origin of the participants is unknown. The authors said "randomly drawn from general population" and "a sample of area residents". Not sure which population they represented?
- I am not sure if there is any significant difference between model 1 and model 2? Why is having a fixed target (e.g., 25% reduction of total cholesterol) more realistic as mentioned in line 88?
- Model 1 and model 2 are personal based approach but model 3 is population-based approach. How can they be directly compared?
- Lines 91-94: This is not what the authors described in the abstract about the minimal prevention strategy?
- Please mention the studied CV risk factors in the abstract.
- All the risk scores used need to be introduced in the introduction. Also, the authors are required to explain why they were chosen.
- Lines 99-119 should be in the result?
- Maybe a flowchart would help to understand the study design, exclusion, inclusion and the number of participants included.